# Exploring the Diversity of Ovule Development in the Novel Rice Mutant *ShuangLi* Using Confocal Laser Scanning Microscopy

**DOI:** 10.3390/plants14192982

**Published:** 2025-09-26

**Authors:** Shuaipeng Zhao, Chunhong Wu, Yuanyuan Hao, Jikun Xu, Jian Li, Qunce Huang

**Affiliations:** 1College of Biological and Pharmaceutical Engineering, Shandong University of Aeronautics, Binzhou 256603, China; zhaoshuaipeng@sdua.edu.cn (S.Z.); wuchunhong.123@163.com (C.W.); xujikun@sdua.edu.cn (J.X.); lijiansdua@sdua.edu.cn (J.L.); 2The Yellow River Delta Sustainable Development Institute of Shandong Province, Dongying 257000, China; xiaoya235@163.com; 3Henan Province Key Laboratory of Ion Beam Bioengineering, Zhengzhou University, Zhengzhou 450052, China

**Keywords:** ovule, embryo sac, laser scanning confocal microscopy, ion beam, rice mutant

## Abstract

Low energy N^+^ ion beam implantation has been used to create the novel rice mutant “*shuangli*”, which produces partially fertile spikelets containing double grains. Abnormal ovule development is a major cause of partial fertility and grain diversity in rice mutants. To elucidate the developmental mechanism of ovule diversity in *shuangli*, ovules undergoing development were stained using eosin Y and H33342 and observed using confocal laser scanning microscopy. Different developmental abnormalities were observed in the ovary, embryo sac, and ovule. Abnormal development was observed in 35.18% of the ovary structures, primarily manifesting as “tumor” like cell clusters, “false ovaries”, stamen degeneration, and double ovaries. In the embryo sac, abnormal development occurred in about 17.35% of the megaspore cells, including the formation of three nuclei, two daughter cells of asynchronously divided dyads, multiple megaspore tetrads, and “narrow and elongated” cavities. At the female gametogenesis stage, the abnormal development rate was 27.53%, mainly involving the degeneration of the central polar nucleus, egg apparatus, antipodal cell mass, or female germ unit. In *shuangli*, abnormal development occurred in 28.06% of the ovule structures, including lateral tissue, nucellar tissue, double ovules and double embryo sacs. Of the observed lateral tissues, 8.27% did not differentiate into sexual reproductive tissue, which affected the fertilization of the embryo sac, leading to atrophy and degeneration. A new abnormal tissue similar to the inner integument was found on both sides of the nucellar tissue, and the two specialized nucellar tissues appeared to have “staggered” growth within a single ovary. Of the examined ovules, 10.79% exhibited different types of double ovules, including heart-shaped, “anatropous”, “conjoined” structures. However, the double ovules typically developed synchronously, explaining the production of different sizes of the two grains in *shuangli*. In addition, “double” embryo sacs from two “twinborn” nucelli were found in one ovule, and the frequency of “double” embryo sacs was 3.60%. Therefore, ovule development diversity may result in fertilization or gradual degeneration after fertilization, explaining the lower fertility of *shuangli* at the embryological level.

## 1. Introduction

Rice produced in China accounts for approximately 36.9% of the world’s total rice production annually. Rice is one of the three major food crops in China, where more than 65% of the population relies on it as their staple food [1]. China’s population is predicted to reach 1.6 billion, and the demand for rice, which is an important foundation of food security strategies, will further increase by 2030 [2]. As the population increases, the rice cultivation environment deteriorates, and the cultivated land area and quality decrease [3]. Therefore, it is very important for rice breeders to continuously explore and innovate to cultivate new rice varieties with high yields, applicability, and resistance [4]. Creating genetically variable materials is the foundation of accelerating breeding speed for the genetic improvement of rice. It plays a crucial role in increasing the mutation frequency, expanding the mutation spectrum, and shortening the screening cycle. Modern biotechnology can be utilized to create various materials, which can provide favorable source materials for the genetic improvement of rice [5,6].

Researchers have primarily created new rice mutants using different biological methods, such as physical irradiation (e.g., ultraviolet light, low-energy ion beams), chemical induction, and molecular design (e.g., chromosome doubling, gene editing) [7,8,9]. Low-energy ion beam implantation, a new irradiation technology for plant genetic improvement and development of plants, has been widely applied in wheat, rice, and maize [10]. It is important in both theory and practice because the range and number of implanted ions can be regulated and controlled, and a high mutation frequency and wide mutation spectrum can be achieved with little damaging force, which can create “ion channels” from the cell wall to the nucleus [11]. Low doses of nitrogen ions can significantly enhance callus induction and differentiation, and inhibit callus browning in autotetraploid rice [12]. The biological effects of ion beam implantation in wheat include notable variations in agronomic traits, including plant height, panicle type, leaf shape, and grain color [13]. Nitrogen ion beams induced a relatively high frequency of variation in cherry, radish, and tomato mainly altering fruit characteristics, plant height, and growth period [14,15]. After implantation, both mitotic division of root tip cells and meiotic division during the gametogenesis phase show abnormalities, including chromosome bridges, free chromosomes, chromosome fragments, lagging chromosomes, dragged chromosomes and abnormal dyads [16]. In addition, ion implantation can induce cell nucleus abnormalities, including binucleation, micronucleation, micronuclei, and nuclear malformation [17].

In recent years, different mutants from common rice and wheat seeds have been generated by γ-ray, UV light, and heavy-ion beam irradiation, and their special biological properties have been studied [18,19]. However, research on mutants has mainly focused on the molecular level, such as gene localization and regulation, and the developmental characteristics of specific mutants, particularly their reproductive properties, have not attracted as much attention. In this study, a novel rice mutant, “*shuangli*” was generated by our team using low-energy N^+^ ion beam implantation. We primarily investigated its morphology and female gametophyte formation and development using laser scanning confocal microscopy. Different types of embryo sacs, female gametophytes, and ovules were observed during ovary development. We also analyzed the relationship between abnormal ovary development and fertility in *shuangli*. The objective was to elucidate the causal mechanism of embryo sac abortion at the cellular level and to reveal how abnormal embryo sacs affect fertility in *shuangli*. Therefore, it is very important that we reveal the cytological formation mechanism of double grains in *shuangli* to supply useful technical support for the molecular design breeding of mutants.

## 2. Results

### 2.1. Abnormal Ovaries in shuangli

To better understand abnormal ovary development in *shuangli*, 468 florets were dissected and analyzed in detail using anatomical microscopy (Motic-SMZ140, Motic China Group Co., Ltd. Xiamen, China). When it was about to enter the flowering stage, abnormal development was observed in the ovary of its pistil. Both normal ovary morphology and various abnormalities were observed during *shuangli* ovary development (Figure 1a). First, the ovaries were abnormally curved and enlarged (Figure 1b). Second, there were “false-ovaries” with abnormally enlarged bases but lacking stigmas around the normal ovaries, as well as there were “tumor”-like cell clusters (Figure 1c). Third, there was a phenomenon in which the stamen degenerated into pistils on one side of the normal ovary (Figure 1d). Fourth, the two ovaries were morphologically normal and of equal size (Figure 1e), or one ovary was fused with an undeveloped lodicule in double ovaries (Figure 1f). As shown in Figure 2, statistical analysis revealed that ovaries with abnormalities accounted for 35.18% of all samples, slightly lower than the 43.26% of normal single ovaries (*p* < 0.05). Normal double ovaries accounted for 11.93%, which is very close to the frequency observed for those without ovary structures, suggesting that abnormal ovary development is an important cause of reduced fertility in *shuangli*.

### 2.2. Formation Process and Characteristics of Embryo Sac Development in CK

From Figure 3, rice embryo sac development goes through nine stages, from sporogenous cell formation to mature embryo sacs. In the sporogenous cell stage, the differentiation of the inner integument began, and the sporogenous cells differentiated from the subepidermal layer near the micropylar end of the nucellus (Figure 3a). The differentiation of the inner integument continued, and the outer integument began to differentiate. The sporogenous cells elongated axially and directly differentiated into megaspore mother cells (Figure 3b). The ovule grew horizontally, and the inner and outer integuments reached the ventral side of the ovary wall but did not penetrate the top of the nucellus from the leptotene to zygotene stages. Passing the pachytene stage, unequal division occurred, and a megaspore dyad formed during telophase of meiosis I (Figure 3c). The megaspore mother cell entered meiosis II (Figure 3d) and produced four megaspores arranged parallel to the longitudinal axis of the ovule (Figure 3e). Subsequently, three of the four megaspores degenerated, sequentially based on their proximity to the micropylar end. Only one megaspore, located at the chalazal end, remained, and it ultimately developed into a functional megaspore (Figure 3f). The functional megaspore elongated and grew, with vacuoles appearing around it, and the size of the vacuoles gradually increased in size and became more easily observable in the mononucleate embryo sac stage (Figure 3g). The nucleus in the mononucleate embryo sac underwent one mitotic division along the longitudinal axis of the embryo sac, forming a binucleate embryo sac, and the middle part of the embryo sac was occupied by a large vacuole (Figure 3h,i). The binucleate embryo sac further developed and underwent one mitotic division to form a tetranucleate embryo sac (Figure 3j), which underwent another synchronous mitotic division to form an octonucleate embryo sac (Figure 3k). When the octonucleate embryo sac entered the maturity stage, the central polar nucleus became highly vacuolated, and the two polar nuclei were closely positioned together, located above the egg apparatus. The egg cell was larger, and the synergid cells were smaller; the antipodal cell mass at the chalazal end was strongly stained and clearly visible (Figure 3l). This reproductive mode and structural arrangement of the embryo sac played a crucial role in facilitating the “double fertilization” process.

### 2.3. Abnormal Embryo Sac Development in shuangli

During embryo sac development in *shuangli*, abnormal changes occurred at different stages. Abnormal development of the megaspore mother cell was observed in 17.35% of the 386 ovary samples, which was more than twice the rate in CK (8.33%; *p* < 0.05) (Table 1). Three nuclei formed instead of a normal dyad at the end of meiosis I in the megaspore mother cell, and the arrangement of these three nuclei was coaxial or non-coaxial (Figure 4a,b). During meiosis II, the two daughter cells of the dyad divided asynchronously. After one had divided into two daughter cells, the other still had not initiated division (Figure 4c). This abnormal development accounted for 5.18% of the 386 *shuangli* ovary samples, which is nearly twice the rate observed in CK (2.56%). During tetrad formation at the end of meiosis II, the four daughter cells were not arranged parallel to the longitudinal axis of the ovule, and one of the daughter cells at the chalazal end was arranged transversely (Figure 4d). This abnormal situation occurred in 3.89% of the samples in *shuangli*, which is two-fold higher than that in CK (1.92%). When the primary megaspore tetrad formed, two additional megaspore tetrads simultaneously appeared on its right side, resulting in multiple megaspore tetrads (Figure 4e). This situation was not observed in the 156 ovary samples in CK, but it occurred in 2.07% of the 386 *shuangli* samples. All four daughter cells of the megaspore tetrad degenerated before reaching the functional megaspore, leaving only a “narrow and elongated” cavity (Figure 4f). This abnormal situation occurred in 3.85 and 6.22% of the CK and *shuangli* samples, respectively.

Compared to CK, the abnormal development rate of 425 ovaries increased more than 2-fold to 27.53% (*p* < 0.01) at the female gametogenesis stage (Table 2), which was the main stage of abnormal development in *shuangli*. In the mature embryo sac, the central cell contained one polar nucleus, with the other polar nucleus degenerating (Figure 4g). Although there were two polar nuclei in the central cell, they did not move to the egg apparatus but stayed near the antipodal cells (Figure 4h). This situation was observed in 6.35% of 425 *shuangli* ovaries, increasing by 67.11% compared to its frequency in CK. Additionally, the antipodal cell mass and central cell were present, but there were no cellular structures at the micropylar end (Figure 4i). This means that the egg apparatus degenerated. This situation was observed in 5.88% of samples, representing an increase of 39.34% compared to that in CK. Additionally, the antipodal cell mass remained in the embryo sac, but the central cell or egg apparatus disappeared, indicating degeneration of the female germ unit (Figure 4j). Abnormal variation was observed in 7.06% of samples, which was nearly twice the rate observed in CK. In normal mature embryo sacs, antipodal cells underwent mitotic and amitotic divisions, and continuously proliferated to form the antipodal cell mass (Figure 4k,l). However, in *shuangli*, some antipodal cells did not form a mass but only exhibited a “dispersed” phenomenon after division (Figure 4m,n), occurring in 4.71% of the sampled population. In addition, the antipodal cell mass, egg apparatus, and the central polar nucleus simultaneously disappeared in mature embryo sacs, and the basic morphology of the embryo sac and some filamentous organelles were observed (Figure 4o,p). A specific scenario occurred in *shuangli* in which the embryo sac structure did not develop in the ovule but instead remained nucellar cells. This variation occurred at frequencies of 1.69 and 3.53% of the sampled populations in CK and *shuangli*, respectively.

**Table 1 plants-14-02982-t001:** Abnormal development in megasporogenesis of *shuangli.*

	Ovary Number	Abnormal Dyads (%)	Abnormal Cell Array of Tetrads (%)	More Tetrad Cells (%)	Tetrad Degeneration (%)	Total Variation (%)
CK	156	2.56 ± 0.73	1.92 ± 0.64	0	3.85 ± 1.12	8.33 ± 1.67
*shuangli*	386	5.18 ± 1.34 *	3.89 ± 1.02 *	2.07 ± 0.53 **	6.22 ± 2.27 *	17.35 ± 3.24 *

Note: The values are expressed as mean ± standard deviation. Asterisks (**) and (*) following the values of *shuangli* indicate significant differences at the 0.05 probability level and highly significant differences at the 0.01 probability level when compared to CK, respectively. During the four developmental stages of megasporogenesis, three groups of replications were established at each stage, with 13 samples collected for CK and approximately 32 samples for *shuangli* in each group.

**Table 2 plants-14-02982-t002:** Abnormal development in female gametophytes of *shuangli.*

	Ovary Number	Abnormal Polar Nuclei Number and Position (%)	Egg Apparatus Degradation (%)	Degradation of Female Reproductive Units (%)	Abnormal Antipodal Cells (%)	Loss or Degradation of the Embryo Sac (%)	Total Variation (%)
CK	225	3.80 ± 1.13	4.22 ± 1.53	3.38 ± 1.37	0	1.69 ± 1.54	13.08 ± 3.28
*shuangli*	425	6.35 ± 1.64 **	5.88 ± 1.62	7.06 ± 2.35 *	4.71 ± 1.82 **	3.53 ± 1.14 *	27.53 ± 4.84 **

Note: The values are expressed as mean ± standard deviation. Asterisks (**) and (*) following the values of *shuangli* indicate significant differences at the 0.05 probability level and highly significant differences at the 0.01 probability level when compared to CK, respectively. During the five developmental stages of female gametophytes, three groups of replications were established at each stage, with 15 samples collected for CK and approximately 29 samples for *shuangli* in each group.

### 2.4. Specific Changes in Ovule Development in shuangli

During normal ovary development, the ovary structure consists of the ovary wall, outer integument, inner integument, nucellar tissue, and embryo sac, from the outer to inner layers. At the maturity stage, the embryo sac was made up of the egg apparatus, central cell, and antipodal cell mass. The egg apparatus was located at the micropylar end, and the antipodal cell mass was at the chalazal end (Figure 5a,b). However, abnormalities also occurred during ovule development. Among 124 ovaries observed in CK, 4.84% of the ovules exhibited abnormal development. Abnormal development occurred in 28.06% of ovule structures in *shuangli*-six times the rate in CK (*p* < 0.01) (Table 3). Based on these observations, abnormalities in the development of the ovule structure were grouped into four major types: abnormal development of lateral tissue in the ovule, abnormal nucellar development, double ovules, and double embryo sacs.

#### 2.4.1. Abnormal Development of Lateral Tissue in the Ovule

During the megasporocyte stage, integument cells near the chalazal end of the embryo sac began to differentiate into lateral tissue located on one side of the sexual reproductive embryo sac (Figure 5c). As the ovule further developed, the integument cells proliferated continuously, and the integument enlarged and thickened (Figure 5d). At the mature stage of the embryo sac, the lateral tissue closely adhered to the inner integument of the embryo sac, and its bottom grew towards the micropylar end, but it did not differentiate into sexual reproductive tissue (Figure 5e,f). Based on the proportion of abnormal differentiation in the glumes observed in the final stages, we do not believe that the lateral tissue developed into the third glume in the spikelet. Instead, we speculate that the apearance of lateral tissue affected the fertilization of the embryo sac, leading to its atrophy and degeneration. Statistical analysis showed that lateral tissue was only found in *shuangli*, where it accounted for 8.27% of the total sampled quantity.

#### 2.4.2. Abnormal Development of Nucellar Tissue

During normal ovule formation, the sporogenous cell differentiated under the nucellar epidermis, and inner and outer integuments formed from both sides of the nucellus. As the sporogenous cell developed into a functional megaspore and female gametophyte, the inner integument grew faster than the outer integument. The inner integument first formed an enclosing structure at the micropylar end during the megaspore stage. However, the outer integument failed to fully enclose the nucellus during female gametophyte formation. During ovule development in *shuangli*, sporogenous cells failed to differentiate at the megasporocyte stage, but the inner and outer integuments differentiated (Figure 5g). As the nucellar tissue continued to grow, the outer integument grew faster than the inner integument, and an enclosing structure formed at the tip of the nucellar tissue (Figure 5h–j). During the further enclosure process of the inner integument at the tip of the nucellus, a new tissue similar to the inner integument (tentatively identified as new inner integument, NII) differentiated on both sides of the nucellar tissue (the inner layer of the inner integument), and there was a trend of enclosure (Figure 5k). As the nucellar tissue developed, the outer integument tissue degenerated, and new integument tissue formed. The entire nucellar tissue continued to rise, approaching the top of the ovary wall (Figure 5l). Simultaneously, a unique phenomenon of “staggered” growth of two specialized nucellar tissues was observed within a single ovary (Figure 5m). Statistical analysis revealed that the specific mode of nucellar differentiation was exclusive to *shuangli*, occurring in 5.40% of the total sample quantity.

#### 2.4.3. Abnormal Formation of Double Ovules

Double-ovule formation is a phenomenon occasionally observed in rice reproductive development, but the types of double ovules in *shuangli* exhibited greater diversity. Based on observations, there were three distinct reproductive patterns. First, upward protrusions on the inner wall of the ovary at the base of the ovarian stalk developed into two ovules that differentiated on either side, forming a “heart-shaped” structure (Figure 5n). During further differentiation, the left ovule developed into an embryo sac, while the right ovule developed into lateral ovule tissue (Figure 5o,p). Second, when the inner wall of the ovary at the base of the ovarian stalk protruded upward, it differentiated into two ovules that grew upward or laterally. Both ovules differentiated into embryo sacs, but these embryo sacs exhibited “anatropous” growth (Figure 5q,r). Additionally, in this scenario, two ovules differentiating from one side of the inner wall in the ovary were also observed, with the same “inverted” phenomenon (Figure 5s). Finally, two ovules were differentiated by the inner ovary wall at the top of the ovary and presented a “conjoined” growth pattern (Figure 5t). With further development, the two ovules were either arranged side by side or one above the other, and both differentiated into embryo sacs or were in the process of differentiating (Figure 5u,v). In other samples, the two ovules differentiated separately from the top of the ovary without exhibiting the “conjoined” phenomenon; the two ovules were approximately equal in size but did not develop synchronously (Figure 5w). Statistical analysis showed that the frequency of double ovules in *shuangli* (10.79%) was nearly seven times higher than that in CK (1.62%).

#### 2.4.4. Abnormal Development of Double Embryo Sacs

The phenomenon of “double” embryo sacs was observed in the ovules of *shuangli* and CK. In one ovule, two “twinborn” nucelli differentiated from the lateral side towards the inner wall of the ovary, subsequently developing into two embryo sacs. The inner and outer integuments of the ovule enclosed each other at the base of the ovary. Both embryo sacs were able to normally form a mature embryo sac structure (Figure 5x). Statistical analysis revealed that the frequency of “double” embryo sacs in *shuangli* was similar to that in CK. These observations revealed that the formation and development of double ovules in *shuangli* exhibited significant abnormalities compared with normal ovule development. The diversity of double ovule types suggests that disruptions or alterations occur in the genetic or hormonal regulation of ovule formation and differentiation in *shuangli*. Further research is needed to investigate the mechanisms and genetic factors responsible for the abnormal developmental pattern.

## 3. Discussion

Ion radiation breeding, a novel mutagenesis technology, has made significant strides in crop improvement. This method involves exposing plant materials to ion beams, which can induce a wide spectrum of mutations and achieve high mutation frequencies, while minimizing physiological damage and ensuring certain directionality and repeatability [20]. Compared to traditional mutagenesis methods, the mutations induced by ion beam irradiation exhibit good stability, high mutation efficiency, and a diverse array of mutation types, making them valuable for creating new germplasm resources. In rice breeding, ion beam irradiation has been effectively utilized to enhance the traits of aromatic rice, resulting in induced mutants that demonstrate stable growth performance and early maturity [21,22].

During reproductive development in rice, normal ovary structure is a prerequisite for “double fertilization”, and it is one of the decisive factors ensuring a normal seed set in rice. Generally, the nucellus tissue located in the ovary, especially the embryo sac structure within the nucellus tissue, is less susceptible to abnormal changes compared to pollen because of the external environment [23]. Therefore, if the ovary structure of rice is normal, the embryo sac generally exhibits high fertility under normal conditions. However, rice mutants induced by ion beam irradiation, especially the floral organ mutants, usually exhibit low fertility and poor seed setting. Researchers studying abnormal development mainly focus on locating the gene changes, while there are few reports on exploring the reasons for sterility from the cytological perspective of spikelet development. In this study, as the experimental material, we selected the floral organ mutant “*shuangli*”, which was induced using ion beam-mediated technology. We investigated the reasons for its low fertility from the perspective of abnormal ovary development using laser scanning confocal microscopy to provide a theoretical basis for a deeper understanding of the cytological characteristics of decreased fertility in mutants.

### 3.1. Abnormal Ovary Development in shuangli

Researchers studying abnormal ovary development in rice have primarily focused on embryo development. The normal rice embryo sac development can be divided into two stages: megasporogenesis and female gametogenesis. Megasporogenesis includes sporogenous cell formation, megaspore mother cell formation, megaspore meiosis, and functional megaspore formation; and female gametogenesis includes mononuclear, two-nucleate, four-nucleate, eight-nucleate, and mature embryo sacs [24,25]. Autotetraploid rice has been intensively used to study embryo sac development. Because of chromosome doubling in autotetraploids, the rate of abnormal embryo sac development is relatively high [26,27,28]. Researchers have observed and analyzed embryo sac development in autotetraploid rice using multiple biological methods such as laser confocal microscopy, scanning electron microscopy, and paraffin sectioning. Studies have revealed various abnormal phenomena, including abnormal chromosome pairing during megasporogenesis, asynchronous cell division, abnormal dyads and tetrad degeneration, egg apparatus and female germ unit degeneration, and changes in the number and position of polar nuclei during the mature embryo sac stage [29,30]. This study focused on the floral organs of the *shuangli* mutant and studied abnormal ovary development from megaspore formation to female gametophyte formation. The types of abnormal embryo sac development in *shuangli* were similar to those in autotetraploid rice, but they possessed unique characteristics. During the tetrad stage, the three tetrads (multiple tetrads) occurred simultaneously in one embryo sac, which is fundamentally different from the abnormal development of a three-nucleate embryo sac in the multi-embryonic rice line APIV and the six-nucleate and nine-nucleate embryo sacs of IR36 [26,28]. At the mature embryo sac stage, both synergid cells located on both the two sides of the egg cell were always observed before flowering in the normal embryo sacs of *shuangli*. This finding differs from the observation that one of the two synergid cells degenerates and disappears very early before flowering, while the other synergid cell (persistent synergid) is always present until flowering [30]. However, this research strongly supports the observational results in diploid rice examined using electron microscopy [31].

Regarding abnormal ovule development, there were four different ovule types in a single ovary in *shuangli*: abnormal development of lateral tissue, nucellar tissue, four types of double ovules, and double embryo sacs. Double ovules were the main abnormal developmental abnormality. Two equivalent double ovules of an ovary have been observed in different genotypes of autotetraploid rice [32]; the same structure also appears in the embryo sacs of *shuangli*, but it is one of four types of abnormal double ovule development. The frequency of double embryo sacs in *shuangli* was approximately 3.6%, which is similar to that in autotetraploid rice but significantly lower than that in polyembryonic rice ApIII (11.3–14.6%) [33]. Regarding the formation of the double embryo sac, one explanation is that two or three (or more) embryo sacs appeared because of abnormal development of synergid cells in rice ovules, but there is no direct evidence to support this speculation in our study. In *shuangli*, we observed that lateral and nucellar tissues could develop abnormally into a double embryo sac; this may be a new mutation type induced by ion beam mediation, and its cytological developmental characteristics need to be further researched.

### 3.2. Fertility and Abnormal Pistil Development in shuangli

For autotetraploid rice, studies have shown that a lower seed setting rate is closely related to the abnormal development of the embryo sac, and the main cause is degeneration of the embryo sac, egg apparatus, and female germ unit without egg cells. The abnormal development of the embryo sac affects normal fertilization and results in sterility [34,35]. Considering the agronomic traits of *shuangli*, the seed setting rate was only about 10%, and most spikelets failed to produce seeds. The reasons for the lower seed setting rate were related to abnormal pistil development. First, the abnormal development of the embryo sac prevented normal fertilization and influenced seed formation, similar to that of autotetraploid rice. Second, abnormal development of the nucellar tissue in the ovule prevented differentiation into an embryo sac, making fertilization impossible. Based on the proportion of abnormal differentiation in the glumes observed in the final stages, we do not believe that the lateral tissue developed into the third glume in the spikelet. Instead, we speculate that the appearance of lateral tissue affected the fertilization of the embryo sac, leading to its atrophy and degeneration. Statistical analysis showed that lateral tissue was only found in *shuangli*, where it accounted for 8.27% (Table 3) of the total sampled quantity. So the abnormal lateral tissue development interfered with the normal development of the ovule and competed for nutrient uptake.

Perhaps the ovule can be fertilized, but it may not successfully differentiate into a grain. The seeds developed from double ovules were not found in the harvested grains, and it was speculated that either both ovules were unfertilized because of competition or they were fertilized but undergoing degeneration because they did not absorb sufficient nutrients during ovary development. Additionally, the double embryo sac may be fertile [36,37], but its frequency during abnormal ovule development in *shuangli* was too small to exhibit a macroscopic effect on the seed setting rate. Abnormal differentiation of the ovary occurred during pistil formation; nearly 45% of the pistils had a normal phenotype (Figure 3e). However, these pistils may encounter abnormal embryo sac or ovule development during the development process, which also affects the seed setting rate of the spikelet. Additionally, approximately 40% of pistils exhibited abnormalities during spikelet development. Based on the final grains, abnormal development may result in fertilization failure in the pistils or they may gradually degenerate after fertilization. This may explain the low fertility of *shuangli* with abnormal pistil development. Of course, there may be other factors affecting the fertility of *shuangli*, such as pollen fertility, fertilization, and development of embryonic tissue and the endosperm, which require further in-depth study.

## 4. Materials and Methods

### 4.1. Materials

In rice genetic improvement experiments, DNA from maize (Xixingchinuo No. 1) was transferred into the recipient material IR36 (CK) (Figure 6a) using ion-beam-mediated technology, resulting in a mutant with “double grains” in the contemporary population [32]. After six generations of isolation, purification, and screening, genetically stable lines of this mutant were obtained. The mutant’s spikelets had multiple glumes, which were long and open. The main agronomic traits of the mutant population were uniform; most of the florets had high sterility (90.27%) (Figure 6b), while a small portion of the florets were self-fertilizing, producing both single (6.52%) (Figure 6c) and double grains (3.21%) (Figure 6d). Based on the grain characteristics, the mutant was named “*shuangli*” [38]. To systematically study the development characteristics of the mutant, individual plant selection was conducted through self-fertilization at the Yellow River Delta Experimental Base of Shandong University of Aeronautics, and the progeny population was tracked and investigated.

### 4.2. Material Fixation

Nuclear fluorescent staining was used to dynamically track embryo sac development, and the whole-ovary clearing technique was performed using a confocal laser scanning microscope [36]. The method was optimized and improved in our experiment. Based on the developmental process of the embryo sacs in the ovaries (megasporogenesis and female gametogenesis), the spikelets were sampled continuously as the following stages: sporogenous, megasporocyte, meiosis, megaspore tetrad, functional megaspore, uninucleate embryo sac, binucleate embryo sac, tetranucleate embryo sac, octonucleate embryo sac, and the mature embryo sac stage. During each sampling, florets in the middle of the spikelets were selected and fixed in formaldehyde-acetic acid-ethanol fixative (FAA) (50% ethanol-glacial acetic acid-30% formaldehyde = 89:5:6). After fixation for 24 h, the samples were rinsed with 70% ethanol and then with double-distilled water (ddw). Under a dissecting microscopy, the ovaries were separated and stored in 70% ethanol for staining and later use.

### 4.3. Eosin Y and Hoechst Staining

During staining, the ovaries were rehydrated with different ethanol concentrations (70, 50, 35, 20, and 15%) and ddw, with every step lasting 30 min. Subsequently, the ovaries were permeabilized with 1 mol/L NaOH for 1 h. When hydrolysis was completed, the ovaries were counterstained using 1 mg/100 mL of eosin Y for approximately 8 h, and then rinsed with ddw until no floating dye was observed. The counterstained ovaries were then placed in 0.1 mol/L citric acid-potassium phosphate buffer (pH = 5.0) for pretreatment for 8 h. Then, they were stained with 35 μg/mL H33342 (Hoechst stain) in the dark at 25 °C for 20 h. After nuclear staining, the ovaries were rinsed three times with ddw, and dehydration was carried out using a gradient series of ethanol solutions (15, 30, 50, 70, 85, and 95%), with each step lasting for 25 min. After dehydration, the ovaries were placed in absolute ethanol twice, for 2.5 h each time, and then left in absolute ethanol overnight. The next day, the dehydrated ovaries were transferred to a mixture of absolute ethanol and methyl salicylate (*V*/*V* = 1:1) for 1 h. Finally, the material was treated with methyl salicylate three times. The first two treatments were separated by intervals of 2 h each, and the final clearing step lasted for more than 15 h.

### 4.4. Scanning and Observation

After fixation for 24 h, spikelet structures at various developmental stages were observed using anatomical microscopy (Motic-SMZ140). The fixed samples were rinsed in a culture dish with distilled deionized water (ddw), and a dissection needle alongside sharp forceps was employed to separate the ovaries from the spikelets. The LED light source of the SMZ140 was activated, and the separated ovaries were placed on a slide or in a Petri dish with a few drops of ddw to prevent desiccation. The target ovary was located using a total magnification of 40 × (10 × eyepiece + 4 × objective lens) to examine the enlarged ovary structure at the base of the flower.

Abnormalities in ovary development were statistically analyzed and photographed using a laser scanning confocal microscopy. To observe the ovaries, cleared ovaries were placed on a concave microscope slide, covered with a coverslip, and sealed with clove oil. The prepared slide was then inverted and placed on the stage of a laser scanning confocal microscopy. The developmental characteristics of various stages were observed at a wave-length of 532 nm, analyzed, and photographed. Composites of four to ten images from different focal planes of a sample are shown. The ovaries used for embryo sac scanning were randomly collected from samples at the same stage.

### 4.5. Statistical Analysis

In the study of embryo sac development, nine distinct stages were identified, with three replicates established for each stage. For each replicate, 15 samples were randomly collected from the CK and 30 samples from *shuangli*, resulting in a total of 1215 samples prepared and analyzed. The investigation focused on the abnormal development of the ovule structure during the mature embryo sac stage. For this analysis, three groups of replications were established, with 41 samples randomly collected from CK and approximately 80 samples from *shuangli* in each replication, leading to a total of 364 samples prepared and analyzed. Additionally, five samples were randomly selected from each spikelet, with three spikelets chosen from each individual rice. Approximately 110 individual rice were utilized in this study.Mutation Rate = (Number of Abnormalities/Number of Samples) × 100%

Graphs were plotted using Origin 2021, and the software SPSS27.0 was used to analyze the significance of differences between *shuangli* and CK by *t*-test. values were expressed as means ± standard deviation. A *p*-value < 0.05 and *p*-value < 0.01 were, respectively, considered statistically significant differences at the 0.05 probability level and highly significant differences at the 0.01 probability level.

## 5. Conclusions

This study systematically investigated the diversity of ovule development in the novel rice mutant “*shuangli*” using confocal laser scanning microscopy. The aim was to elucidate various abnormal developmental patterns across different stages and structures that contribute to its partial fertility and double-grain phenotype. The results revealed pervasive abnormal development in ovarian structures, embryo sac formation, and ovule morphology, with distinct abnormalities identified at key stages of female reproductive development. Notable abnormalities in the ovary included “tumor-like” cell clusters, “false-ovaries”, stamen degeneration, and double ovaries. Furthermore, embryo sac development exhibited stage-specific defects: 17.35% of abnormalities were observed at the megaspore cell stage (e.g., formation of three nuclei, asynchronous dyad division, multiple megaspore tetrads, and “narrow and elongated” cavities), while 27.53% were noted during female gametogenesis (involving degeneration of central polar nuclei, egg apparatus, antipodal cells, and female germ units). Ovule structural abnormalities were characterized by defects in lateral tissues, anomalies in nucellar tissue (including novel inner integument-like lateral tissues and “staggered” growth of two specialized nucelli), and a variety of double-ovule phenotypes, such as “heart-shaped”, “anatropous”, and “conjoined” structures, which typically developed synchronously and likely underlie the production of two grains of different sizes. “Double embryo sacs” derived from “twinborn” nucelli were observed within single ovules. These findings collectively suggest that the diversity of ovule development in “*shuangli*”, including abnormal ovarian differentiation, stage-specific embryo sac defects, lateral tissue differentiation failures, nucellar structure variations, and the formation of double ovules/embryo sacs, directly influences fertilization success and subsequent post-fertilization development. This leads to either partial fertilization or gradual degeneration of reproductive structures, explaining the mutant’s reduced fertility at the embryological level. The discovery of novel abnormalities, such as inner integument-like lateral tissues and “staggered” nucellar growth, offers new insights into the regulatory mechanisms governing the diversity of ovule development in rice. In addition, the synchronous development of double ovules provides a mechanistic explanation for the mutant’s distinctive double-grain trait.

## Figures and Tables

**Figure 1 plants-14-02982-f001:**
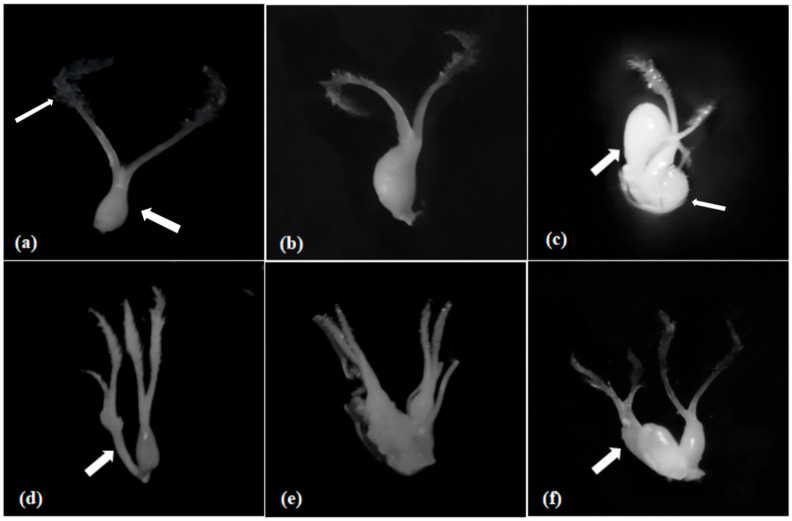
Abnormal ovary variations. (**a**) Normal ovary, with the thick arrow and the thin arrow, respectively, representing the ovary and the stigma. (**b**) Enlarged ovary. (**c**) Presence of “pseudo-ovaries” and “tumor-like” growths around the ovary, with the thick arrow indicating the enclosed ovary and the thin arrow indicating the “false-ovaries”. (**d**) The androgyny phenomenon occurring next to a normal ovary, with the arrow indicating the pistilloid-stamen. (**e**) Two normal double-ovary structures. (**f**) A double-ovary structure, with one fused with a lodicule, and The arrow represents the ovary fused with lodicule.

**Figure 2 plants-14-02982-f002:**
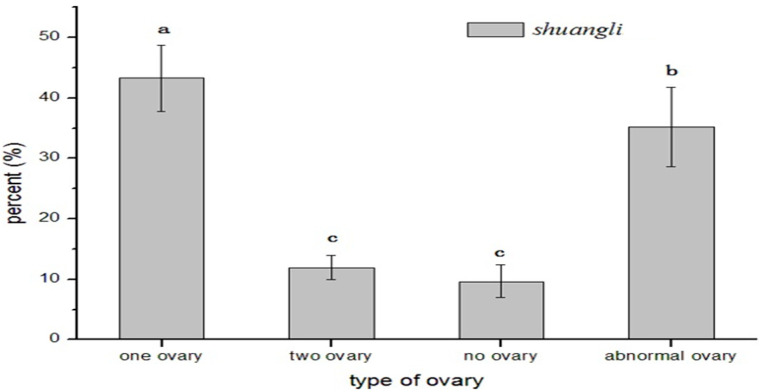
Difference in ovary type in *shuangli.* Note: The columns represent the percent of four ovary types in the statistical results from three groups of repeated experiments, and each group involved 156 observation samples at maturation stage of the ovary. Different lowercase letters above a column in samples indicate significant difference among ovary types at the 0.05 probability level.

**Figure 3 plants-14-02982-f003:**
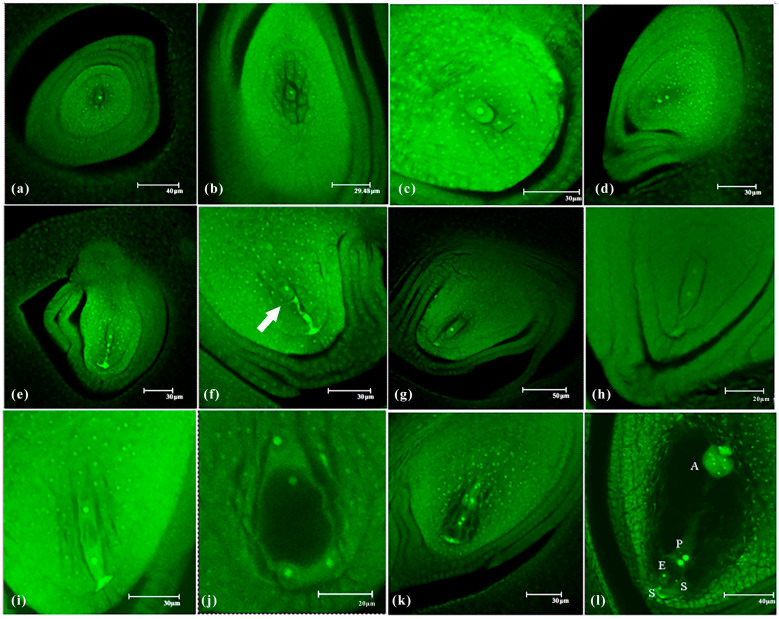
Embryo sac development process in *shuangli*. (**a**) Sporogenous cell. (**b**) Megasporocyte. (**c**) Telophase of meiosis I (formation of the megaspore dyad). (**d**) Prophase of meiosis. (**e**) Megaspore tetrad. (**f**) Functional megaspore. The thick arrow indicated functional megaspore. (**g**) Uninucleate embryo sac stage. (**h**) Early binucleate embryo sac stage. (**i**) Mid-binucleate embryo sac stage. (**j**) Tetranucleate embryo sac stage. (**k**) Octonucleate embryo sac stage. (**l**) Mature embryo sac stage. Note: In these images, S represents synergids, E represents the egg cell, P represents polar nuclei, and A represents antipodal cells. Among the nine stages of embryo sac development in CK, three replicates were established for each stage, with 15 samples collected per replicate. A total of 405 samples were analyzed, with 10 to 20 images captured for each sample using laser confocal microscopy. The most representative images were selected for each stage.

**Figure 4 plants-14-02982-f004:**
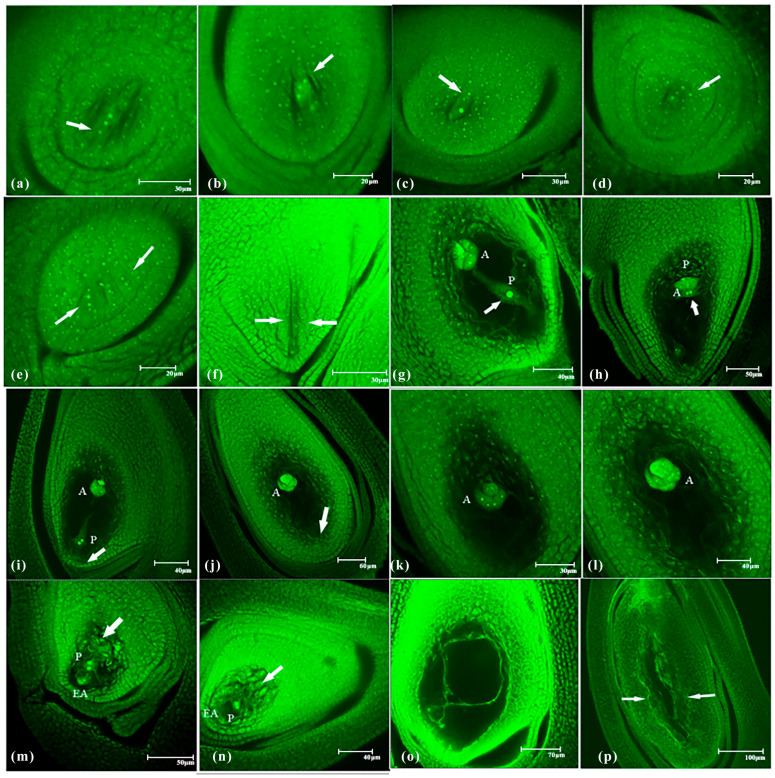
Abnormal embryo sac development in *shuangli*. (**a**) The three cells are aligned longitudinally parallel to the embryo sac, with the arrow indicating the third smaller cell nucleus. (**b**) The three cells are not aligned in the same direction, with the arrow indicating the third smaller cell. (**c**) During metaphase of meiosis II, the two daughter cells of the megaspore dyad split asynchronously, with the arrow indicating the daughter cells that have completed division at the junction end. (**d**) The megaspore tetrad forms; the four daughter cells are not aligned longitudinally parallel to the embryo sac at the end of meiosis II, with the arrow indicating the daughter cells whose division direction is perpendicular to the axial direction of the embryo sac. (**e**) Multiple megaspore tetrads form, with the arrows indicating two other large spore tetrals. (**f**) The tetrad cells undergo complete degeneration, with the arrow indicating the absence of tetrad cells. (**g**) One of the central polar nuclei undergoes degeneration, with the arrow indicating a polar core. (**h**) The central polar nucleus is close to the antipodal cell mass, with the arrow indicatingthe position of the central polar nucleus. (**i**) The egg apparatus in the embryo sac undergoes degeneration, leaving only the antipodal cell mass and the central polar nucleus, and there are no oocytes indicated by the arrow. (**j**) The female germ unit undergoes degeneration, leaving only the antipodal cell mass in the embryo sac, with the arrow indicating a non-magnetic reproductive unit. (**k**) Three antipodal cells at the chalazal end have just undergone mitotic division. (**l**) The antipodal cells form an antipodal cell mass. (**m**,**n**) The antipodal cells have developed but have not formed an antipodal cell mass; instead, they appear dispersed at the chalazal end. The two arrows indicate antipodocytes are scattered at the junction ends. (**o**,**p**) The antipodal cell mass, egg apparatus, and central polar nucleus simultaneously disappear in mature embryo sacs. The arrows indicate the blastocyst morphology exists, but there is no 7-cell 8-nucleus structure. Note: In these images, S represents synergids, E represents the egg cell, P represents polar nuclei, and A represents antipodal cells. Among the nine stages of embryo sac development in *shuangli*, three replicates were established for each stage, with 30 samples collected per replicate. A total of 810 samples were analyzed, with 10 to 20 images captured for each sample using laser confocal microscopy. The most representative images were selected for each stage.

**Figure 5 plants-14-02982-f005:**
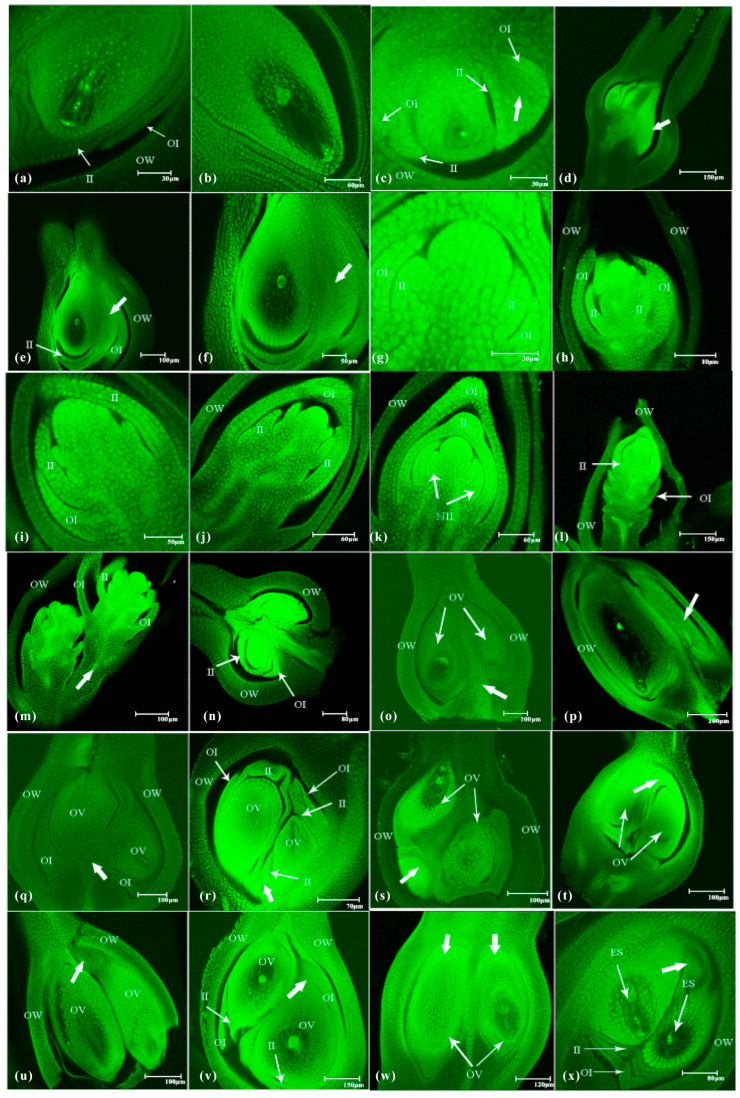
Abnormal reproduction mode in *shuangli* ovules. (**a**) The overall outline of a normal embryo sac. (**b**) Partial enlargement of a normal embryo sac. (**c**) Lateral tissues differentiate on the right side of the nucellus at the megasporocyte stage, with the thick arrow indicating the lateral tissue. (**d**) Overall longitudinal outline of the pistil, with the thick arrow showing lateral tissue development. (**e**,**f**) Overall outline of the ovary at the mature embryo sac stage, with the thick arrow indicating lateral tissues. (**g**) Nucellus cells differentiate into inner and outer integuments. (**h**) Inner and outer integuments appear, and the outer integuments grow faster than the inner integuments. (**i**) Nucellus cells continue to proliferate and develop, and the inner and outer integuments further enclose each other. (**j**) Outer integuments on both sides of the ovule enclose the apex of the nucellus tissue earlier than the inner integument. (**k**) Inner and outer integuments enclose the nucellus, and new tissue similar to the inner integument appears on the sides of the nucellus tissue. (**l**) The outline of abnormally developed nucellus tissue in the ovary. (**m**) the outline of two “staggered” nucellus structures developed in the ovary; the thick arrow shows an abnormal lateral ovule. (**n**) Two “heart-shaped” ovules form due to upward protrusions of the ovary wall at the base of the ovary. (**o**) Development of the two ovules is asynchronous, with the thick arrow indicating the insertion position. (**p**) One of the two ovules develops into a structure similar to the lateral ovule tissues, with the thick arrow indicating the lateral tissue. (**q**) Two ovules attached to the base of the ovary, with the thick arrow indicating the insertion position. (**r**) Two ovules show the “retrograde” growth phenomenon at the mature embryo sac stage; the thick arrow indicates the attachment to the base of the ovary. (**s**) A “retrograde” growth phenomenon in double ovules attached near the left ovary wall, and the thick arrow indicates the insertion position closing to the left ovary wall.(**t**) Double ovules with a “twinborn” phenomenon differentiating the top of the ovary; the thick arrow indicates the connection point between the two ovules. (**u**) “Twinborn” double ovules arranged side by side within the ovary; the thick arrow indicates the position of the connection. (**v**) “Twinborn” double ovules arranged vertically within the ovary, the thick arrow indicates the position of the connection. (**w**) Two independent ovules differentiating at the top of the ovary; the thick arrow indicates the respective attachment positions of the two ovules. (**x**) A “double embryo sac” ovule attached to the left inner wall of the ovary; the thick arrow indicates the connection point between the two embryo sacs. Note: In these images, OV represents ovule, II represents inner integument, OI represents outer integument; OW represents ovary wall; ES represents embryo sac. In this figure, all images of the ovary were captured with the stigma pointing upward.

**Figure 6 plants-14-02982-f006:**
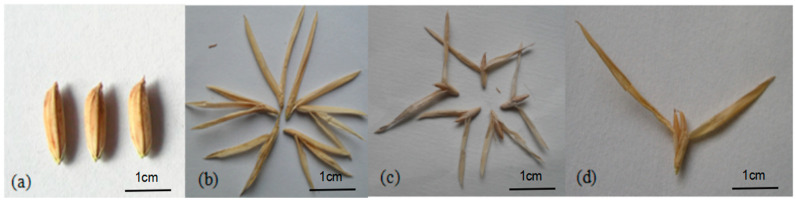
Different grain types. (**a**) Normal grains of IR36 (CK). (**b**) Sterile grains of *shuangli*. (**c**) Single grains of *shuangli*. (**d**) Double grains of *shuangli.*

**Table 3 plants-14-02982-t003:** Abnormal development of ovules at the mature embryo sac stage in *shuangli.*

	Ovary Number	Normal Ovule (%)	Abnormal Development of Ovule Structure				
			Lateral Tissue (%)	Abnormal Nucellar (%)	Double Ovules (%)	Double Embryo Sacs (%)	Total Variation (%)
CK	124	95.16 ± 5.68	0	0	1.62 ± 0.61	3.23 ± 1.25	4.84 ± 1.37
*shuangli*	240	71.94 ± 3.81 *	8.27 ± 2.26 **	5.40 ± 1.76 **	10.79 ± 3.44 **	3.60 ± 1.64	28.06 ± 4.52 **

Note: The values are expressed as mean ± standard deviation. Asterisks (**) and (*) following the values of *shuangli* indicate significant differences at the 0.05 probability level and highly significant differences at the 0.01 probability level when compared to CK, respectively. The abnormal development of the ovule structure at the mature embryo sac stage was investigated. Three groups of replications were established, with 41 samples collected for CK and approximately 80 samples for the *shuangli* group in each replication.

## Data Availability

The original contributions presented in this study are included in the article. Further inquiries can be directed to the corresponding author.

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
