# Peer review of "Exploring the Diversity of Ovule Development in the Novel Rice Mutant ShuangLi Using Confocal Laser Scanning Microscopy"

_plants, 2025, doi:10.3390/plants14192982_

Round 1

Reviewer 1 Report

Comments and Suggestions for Authors

Line 92. The florets of shuangli were dissected and analyzed. This comment is for MyM.

Point out in figure 1a: ovary, stigma, stamen

Line 96. There are “false-ovaries” with abnormally enlarged bases but lacking stigmas around the normal ovaries. Point out!.

Line 101. To better understand the abnormal development of ovaries in shuangli, detailed dissection and analysis were conducted on 468 florets. This comment is for MyM.

Line 145. The antipodal cell mass at the chalazal end was deeply stained and clearly visible (Figure 3l). Can the number of antipodes be quantified?

Line 151. Figure 3 (f) Functional megaspore. Point with arrow.

Fig 6 Add measure.

Line 448. from booting to flowering stage. You should be more precise in defining the phases. Is flowering anthesis?

Why does the discussion not include results obtained with other authors applying the same mutagenesis technique? for example: Phanchaisri, B., Chandet, R., Yu, L. D., Vilaithong, T., Jamjod, S., & Anuntalabhochai, S. (2007). Low-energy ion beam-induced mutation in Thai jasmine rice (Oryza sativa L. cv. KDML 105). Surface and Coatings Technology201(19-20), 8024-8028.

Reviewer 2 Report

Comments and Suggestions for Authors

This manuscript presents the results of research on the diversity of ovule development in a new mutant of shuangali rice. The results presented in the paper are interesting and quite well-documented, contributing to a better understanding of the causes of the mutant's sterility and describing how developmental abnormalities occur in the mutant at the microscopic level.

The paper is interesting and certainly worth publishing. However, several elements require improvement, clarification, or clarification.

  1. Description of Fig. 2 - explanations of how the measurements and calculations were performed are missing, the letter symbols on the graph are not explained in the description, and information on statistical analysis is missing. The description of the y-axis is imprecise - % of what?
  2. Lines 117-121 - are more suitable for discussion. It is better not to place references in the results section. They should be placed in the "discussion" section.
  3. Fig. 3 - How many slides were prepared and analyzed? Does the figure contain the most typical images? This is not explained in the figure description.
  4. Table 2 and Table 3 descriptions are incorrectly placed (Table 2). They lack information on how many replicates statistical significance was calculated from, and the meaning of the asterisk and +/- is not explained.

    5. Fig. 4 description - Please indicate how many slides were prepared and analyzed.

  1. Lines 251-256 - these are more relevant to the discussion.
  2. Lines 255-256: “Statistically, lateral tissue was only found in shuangli, accounting for 8.27%” - on what basis was this calculated? The same applies to data from lines 277 (5.40%), 298 (10.79% and 1.62%), and 305 (11.46%).
  3. Chapter 4.1 Materials (lines 425-438) including Fig. 6: If the mutant characteristics presented there have been previously published, please provide a reference. If this data is previously unpublished, it should be included in the results section.
  4. Fertility and abnormal pistil development of shuangli – how many replicates of the experiment were performed, how many samples were analyzed, and from how many individuals?
  5. Statistical analysis – line 483 – how many samples were in each replicate? Were these technical or biological replicates?

Round 2

Reviewer 2 Report

Comments and Suggestions for Authors

Accept in present form